# ^1^H NMR Profiling of Honey Bee Bodies Revealed Metabolic Differences between Summer and Winter Bees

**DOI:** 10.3390/insects13020193

**Published:** 2022-02-12

**Authors:** Saetbyeol Lee, Filip Kalcic, Iola F. Duarte, Dalibor Titera, Martin Kamler, Pavel Mrna, Pavel Hyrsl, Jiri Danihlik, Pavel Dobes, Martin Kunc, Anna Pudlo, Jaroslav Havlik

**Affiliations:** 1Department of Food Science, Faculty of Agrobiology, Food and Natural Resources, Czech University of Life Sciences Prague, Kamycka 129, 165 00 Prague, Czech Republic; lees@af.czu.cz (S.L.); filip.kalcic@gmail.com (F.K.); mrna.pavel@gmail.com (P.M.); anna.pudlo@upwr.edu.pl (A.P.); 2Campus de Santiago, University of Aveiro (CICECO), 3810-19 Aveiro, Portugal; ioladuarte@ua.pt; 3Department of Zoology and Fisheries, Faculty of Agrobiology, Food and Natural Resources, Czech University of Life Sciences Prague, Kamycka 129, 165 00 Prague, Czech Republic; titera@af.czu.cz; 4Honeybee Research Institute, Dol 94, 252 66 Maslovice, Czech Republic; beedol@beedol.cz; 5Department of Experimental Biology, Faculty of Science, Masaryk University, Kamenice 5, 625 00 Brno, Czech Republic; hyrsl@sci.muni.cz (P.H.); pavel.dobes@mail.muni.cz (P.D.); martin.kunc@sci.muni.cz (M.K.); 6Department of Biochemistry, Faculty of Science, Palacky University Olomouc, Slechtitelu 27, 783 71 Olomouc, Czech Republic; jiri.danihlik@upol.cz

**Keywords:** *Apis mellifera*, winter bees, nuclear magnetic resonance, metabolome, longevity

## Abstract

**Simple Summary:**

The European honey bee, *Apis mellifera*, is well-known to have two distinct populations in temperate climate zone: short-living summer bees and long-living winter bees. Several biological factors related to the different lifespans of the two populations have been studied. However, the link between the metabolic changes and basic physiological features in the bodies of summer bees and winter bees is limited. This study aimed to identify the metabolic fingerprints that characterize summer and winter bees using proton nuclear magnetic resonance (^1^H NMR) spectroscopy. In total, we found 28 significantly changed metabolites between the two populations. The results suggest that the metabolites detected in honey bee bodies can distinguish the summer and winter bees. Changes in carbohydrates, amino acids, choline-containing compounds, and an unknown compound were noticeable during the transition from summer bees to winter bees. The results from this study give us a broad perspective on honey bee metabolism that will support future research related to honey bee lifespan and overwintering management.

**Abstract:**

In temperate climates, honey bee workers of the species *Apis mellifera* have different lifespans depending on the seasonal phenotype: summer bees (short lifespan) and winter bees (long lifespan). Many studies have revealed the biochemical parameters involved in the lifespan differentiation of summer and winter bees. However, comprehensive information regarding the metabolic changes occurring in their bodies between the two is limited. This study used proton nuclear magnetic resonance (^1^H NMR) spectroscopy to analyze the metabolic differences between summer and winter bees of the same age. The multivariate analysis showed that summer and winter bees could be distinguished based on their metabolic profiles. Among the 36 metabolites found, 28 metabolites have displayed significant changes from summer to winter bees. Compared to summer bees, trehalose in winter bees showed 1.9 times higher concentration, and all amino acids except for proline and alanine showed decreased patterns. We have also detected an unknown compound, with a CH_3_ singlet at 2.83 ppm, which is a potential biomarker that is about 13 times higher in summer bees. Our results show that the metabolites in summer and winter bees have distinctive characteristics; this information could provide new insights and support further studies on honey bee longevity and overwintering.

## 1. Introduction

The European honey bee (*Apis mellifera*) is one of the most important pollinators for agriculture and helps support ecological diversity [1]. However, the steadily reported decline in honey bee populations’ ability to overwinter in various regions, including North America and Europe, is a worrisome phenomenon [2,3,4,5]. The elevated overwintering mortality as observed in the last few decades [6] is probably caused by several biotic and abiotic stressors, such as *Varroa destructor* [7], *Nosema* sp. [8], viruses [9], pesticides [10], diet and nutrition [11], poor winter conditions [12], and climate change [13]. As winter is a critical period for honey bee colonies, understanding the positive and negative environmental factors that influence colony mortality will be a crucial task to saving honey bees.

In temperate climates, European honey bee workers emerge into roles as either short-living summer bee or long-living winter bee populations, depending on the emergence period and labor division within the colony. Summer bees mainly take care of their brood, collect nectar, pollen, and water, and have a short life span which lasts an average of 15 to 38 days [14]. When brood rearing begins to cease in the autumn, the winter bees begin to emerge [15,16]. Their main tasks are to keep the colony warm throughout the winter and rear the first summer generation around late winter or in early spring [17], and live approximately 140 and up to 320 days [14]. 

The differences in longevity are linked to several physiological and biochemical characteristics. The protein vitellogenin, juvenile hormone, and their interconnections are one of the most well-known mechanisms that influence the aging process in honey bees. Winter bees have low levels of juvenile hormone along with high levels of vitellogenin [18]; this is similar to queen bees, who have the longest lifespan among the colony members [19]. However, summer bees show the opposite pattern, high levels of juvenile hormone and low levels of vitellogenin [18]. Vitellogenin appears to play a role that affects the aging and lifespan of honey bees in beneficial ways such as its antioxidant function, its role in immunity, and controlling inflammation [20].

Apart from the differences in vitellogenin and juvenile hormone, winter bees have a greater dry weight and a higher amount of total proteins, glucose, glycogen, and lipids compared to summer bees [21,22]. Additionally, they have higher antibacterial activity [22] and different gut microbiota profiles, which appear to be affected by dietary differences in comparison to summer bees [23]. Moreover, winter bees showed a negligible level of carbonylation damage in the brain [24] and negligible cognitive senescence [25] when compared to summer foragers. 

Although many studies have explained the characteristics of winter bees, their biochemical pathways and associated mechanisms have not been fully elucidated. Hence, screening the biomarkers that link the physiological factors that underlie aging and lifespan in honey bees is important considering that the emergence of the long-living population in late summer is crucial for successful overwintering. That could infer the diagnostic biomarkers that are potentially influenced by both positive and negative environmental factors.

Metabolomics entails the comprehensive profiling of metabolites in biofluids, cells, and tissues, which potentially enables biomarker discovery and can provide new insights into the mechanisms that determine various physiological conditions [26]. Nuclear magnetic resonance (NMR) offers an exquisitely informative way to obtain a holistic quantitative picture of small molecules [27]. This approach has been employed in the field of entomology, analyzing fruit flies [28], locusts [29], parasitoid wasps [30], aphids [31], and recently in honey bees [32].

The aim of this study is to apply NMR spectroscopy-based metabolomics to identify the metabolic fingerprints that characterize summer and winter bees of the same age in the temperate zone of the Northern Hemisphere. The period of sample collecting, late June and late August for summer and winter bees, respectively, ensures a consistency in the age of bees. This allows for an unbiased comparison of the transition of the metabolites from short-living populations to long-living populations. The results are intended to provide additional information that links previous research regarding the lifespan of honey bees and to propose potential indicator metabolites necessary for successful overwintering in honey bee colonies.

## 2. Materials and Methods

### 2.1. Sample Collection

Honey bees (*Apis mellifera carnica*) were collected in 2017 over two days, 27th June and 29th August, from three hives at the Honeybee Research Institute, Kyvalka, Czech Republic (49°11′28.7″ N 16°26′58.4″ E). The honey bees were kept under typical bee management conditions and fed sucrose solution (3:2 *w*/*v* sucrose:water) for overwintering on August 8th, 19th, and 24th; no other supplements were added to the hives. The colonies did not show any clinical symptoms of diseases or *Varroa* infestation at the time of sampling. Samples were collected as a subset of another study published earlier [33]. The first sample collection in June represented normal brood rearing activities, whereas the second sample collection in August showed reduced brood activities, which is a sign that the colony is entering the winter bee stage. 

To prepare the same age of honey bees, the method from [34] was applied. Three isolated frames (39 × 24 cm) containing capped brood that will emerge in one to three days were placed in frame cages and kept in three hives. Newly emerged bees were collected from each hive, marked with a color on their thoraxes, then returned to their respective hives. After 10 days of living in the hives, the marked bees were re-collected. The re-collected honey bees were transferred to a laboratory and kept in cages at 30 °C for 24 h on 60% sucrose solution to remove the bias of short-term dietary influences. Thus, the total age of the bees was 11 days. Upon collection, whole honey bees were stored at −80 °C until the time of the analysis. Previously, we have demonstrated on the samples from the same sampling that honey bees collected in August showed significantly higher concentrations of vitellogenin and vitellogenin gene expression (*p* < 0.005) compared with the honey bees collected in June [33]. The color-marked honey bees from August continued to be found in the hive until February. Therefore, this study considers these samples as winter bee population.

### 2.2. Sample Preparation for NMR Analysis

Only the body (head, thorax with legs and wings; and abdomen) was used for NMR analysis. Venom sacks and gastrointestinal tracts of the bees were removed with tweezers and excluded. Honey bee bodies were individually homogenized in a 2.0 mL Eppendorf tube by freezing with liquid nitrogen and then ground to a fine powder by a Dremel MultiTool with an attached burr (Dremel, Mount Prospect, WI, USA). The total number of samples included in the analysis was 120. June samples were 75 (25 per hive) and August samples were 45 (15 per hive). Homogenized bodies were extracted with 1.2 mL of methanol in an ultrasonic bath for five minutes, centrifuged at 14,000× *g* for five minutes, and evaporated in a centrifugal vacuum concentrator for two hours (MiVac Duo, Genevac, Ipswitch, UK). After evaporation, the sample was re-suspended in 540 µL of D_2_O. Subsequently, 60 µL of NMR solution (1.5 M K_2_HPO_4_/NaH_2_PO_4_ pH 7.4 in D_2_O, 5 mM Trimethylsilylpropanoic acid, and 0.2% NaN_3_) was added and vortexed, and the sample was transferred into NMR tubes (5 mm, 7”, High-Throughput, SP Wilmad-Labglass, Wineland, NJ, USA). 

### 2.3. ^1^H NMR Analysis

Standard 1D ^1^H NMR spectra (‘noesypr1d’ Bruker pulse program) of honey bee methanol extracts reconstituted in D_2_O were acquired on a Bruker Avance III spectrometer (Bruker, Billerica, MA, USA) operating at 500.23 MHz for ^1^H observation, at 298 K, and using a 5 mm Broadband obseve probe. The spectra were acquired with 4.9 s acquisition time, 1 s relaxation delay (D1), 0.1 s mixing time, 6510.42 Hz spectral width, 64 K data points, and 64 scans. 

### 2.4. Data Processing and Statistical Analysis

All spectra were zero-filled to 128 K, exponentially multiplied (line broadening 0.3 Hz), manually phased, and baseline-corrected using Topspin 3.5 (Bruker, Billerica, MA, US). For metabolite identification and quantification, we used Chenomx NMR Suite 8.6 (Chenomx, Edmonton, AB, Canada) comprising the Reference Library 10 and our in-house library of approx. 80 compounds. The additional process for identifying metabolites was confirmed through the standard compound analysis (Sigma-Aldrich, St. Louis, MO, USA) and VWR International, Stribrna Skalice, Czech Republic). A set of 36 metabolites was fitted to the spectra and data were manually curated. Compounds below the limit of detection were replaced with one-third of the minimum concentration detected in the dataset. Metabolite concentrations were expressed in µg per bee. The average weight of a bee was 157 mg, of which 48 mg was a digestive tract that was not used for analysis. 

This study used R ver.4.1.2 (RStudio, Boston, MA, USA) for the multivariate analysis. Data were processed using probabilistic quotient normalization, log transformation, and Pareto scaling by the package ‘MetaboAnalystR’ [35]. Orthogonal partial least squares-discriminant analysis (OPLS-DA) with permutation test validation was applied in MetaboAnalyst 4.0 [36]. The packages ‘FactoMineR’ [37] and ‘ggplot2′ [38] were used for the principal component analysis and its visualization.

An independent *t*-test was used to compare the concentration of metabolites in summer bees and winter bees using SPSS statistics ver. 26 (IBM, Armonk, NY, USA), and those metabolites with *p* < 0.05 after Benjamini–Hochberg false discovery rate correction were considered statistically significant. Comparative visualizations for summer and winter bees were performed using the package ‘ggplot2′ [38] in R.

## 3. Results

### 3.1. NMR Spectrum of Honey Bee’s Body

A total of 36 metabolites were identified in the NMR spectra of the honey bee methanolic extracts: 13 amino acids, 4 carboxylic acids, 5 amines, 3 choline-containing compounds, 3 nucleosides, 2 nucleotides, 2 monosaccharides, 2 disaccharides, 1 nucleobase, and 1 unknown compound.

Figure 1 shows a typical ^1^H NMR spectrum of honey bees’ body with the expansion of the signal between the 6.0 and 9.0 ppm region. The region from 0.5 to 3.5 ppm contains peaks mainly from amino acids, organic acids, and lipids. The region between 3.5 and 5.5 ppm contains resonances of mainly carbohydrates, and the region beyond 6.0 ppm contain mainly aromatic compounds.

### 3.2. Multivariate Analysis Explaining the Discrimination between Summer (June) and Winter Bees (Late August)

This study posed a major question: Can summer and winter bees be discriminated by metabolites? If so, what are the key metabolites that could separate these two groups? To answer the question, we used sample set of worker bees comprised of 75 summer bees (June) and 45 winter bees (late August) both of equal age of 11 days.

In this study, the principal component analysis (PCA) showed the separation between summer and winter bees, and the correlated variables that contribute to each of the principal components (Figure 2). The first two principal components explain 54.7% of the total variation in the data (PC1 42.6%, PC2 12.1%) and have an eigenvalue > 1 in PC1. Summer and winter bees showed a notable separation where summer bees formed a cluster to about half of the PC1 whereas, winter bees more tightly clustered over the PC2 than summer bees (Figure 2a). 

The significant variables which separate the summer and winter bees are presented in the variable correlation plot (Figure 2b). The arrow represents the amount of variance positively or negatively correlated on the principal component. The color and length refers the contribution of variables on the principal component in percentage [37]. Among the top five distinguished variables on the PC1, an unknown compound had the highest contribution at 11.6%, followed by sucrose, phenylalanine, trimethylamine N-oxide, and leucine (10.7%, 7.2%, 6.9%, and 6.3%, respectively). Fructose, sucrose, and glucose were the majority of variables on the PC2 which contributed 23.2%, 20.7%, and 17.9%, respectively (Figure 2b, Appendix A). Taken together, the unknown compound, sucrose, fructose, glucose, phenylalanine, trimethylamine *N*-oxide, and leucine were the meaningful variables which contributed to the separation between summer and winter bees. In addition, the permutation test from the orthogonal partial least squares-discriminant analysis (OPLS-DA) confirmed the high validity at Q^2^ = 0.908, R^2^Y = 0.946 when calculated from the 1000 different model permutations.

### 3.3. Univariate Analysis Confirmed Statistically Significant Metabolites

After evaluating the link between the honey bees in summer and winter with certain metabolites by PCA (Figure 2), an independent *t*-test with Benjamini–Hochberg correction was applied to demonstrate the changes in concentrations of individual metabolites between summer and winter bees. From the 36 metabolites, 28 were significantly different between these two groups (Figure 3, Table 1).

The concentration of all the carbohydrates (fructose *p* = 0.0001, glucose *p* > 0.05, sucrose *p* < 0.0001, trehalose *p* < 0.0001, Figure 3a) increased in winter bees compared to summer bees. For example, trehalose in winter bees (801.8 ± 212.0 μg/bee, Table 1) showed 1.9 times higher concentration than in summer bees (416.0 ± 181.1 μg/bee, Table 1).

In the amino acids group, all amino acids (Asparagine *p* < 0.0001, glutamine *p* = 0.0001, isoleucine *p* < 0.0001, leucine *p* < 0.0001, lysine *p* < 0.0001, phenylalanine *p* < 0.0001, threonine *p* < 0.0001, and valine *p* < 0.0001, Figure 3b) except for alanine and proline showed a noticeable decrease in winter bees. Phenylalanine showed the largest difference, with levels 3.4 times higher in summer bees (4.39 ± 2.53 μg/bee, Table 1) than in winter bees (1.31 ± 0.55 μg/bee, Table 1). Alanine (*p* = 0.0001, Figure 3b) and proline (*p* < 0.0001, Figure 3b) were significantly increased in winter bees compared to summer bees.

For the choline-containing compounds, *O*-phosphocholine and choline presented opposite trends (Figure 3c). The level of *O*-phosphocholine showed a statistically significant increase in winter bees (*p* < 0.0001, Figure 3c), while the levels of choline showed a statistically significant decrease in winter bees (*p* < 0.0001, Figure 3c).

The group of nucleobases, nucleotides, and nucleosides are presented in Figure 3d. AMP, adenosine, and NAD^+^ showed a statistically significant increase in winter bees compared to summer bees (all *p*-value < 0.0001, Figure 3d) while adenine, inosine, and uridine decreased (Adenine: *p* < 0.0001, inosine: *p* < 0.0001, uridine: *p* = 0.0499, Figure 3d). Adenine showed the largest differences between the groups at 2.8 times higher in summer bees (1.29 ± 0.82 μg/bee, Table 1) than in winter bees (0.46 ± 0.25 μg/bee, Table 1).

The rest of the metabolites, such as carboxylic acids and amines, are presented as a group in Figure 3e. Trimethylamine *N*-oxide showed notable differences between the two groups, where the concentration in winter bees (2.97 ± 1.54 μg/bee, Table 1) was 3.5 times higher than in summer bees (0.86 ± 0.63 μg/bee, Table 1). Most noteworthy is that the unknown compound, a methyl singlet at 2.83 ppm, showed the greatest change among all metabolites (*p* < 0.0001, Figure 3e), which was about 13 times more concentrated in summer bees (19.61 ± 7.62 μg/bee, Table 1) compared to winter bees (1.53 ± 0.55 μg/bee, Table 1).

## 4. Discussion

This study utilized NMR metabolomics to investigate whether summer bees and winter bees could be distinguished by specific metabolites and identified that 28 metabolites significantly changed during the transition from summer to winter bees. Among the 28 metabolites distinguishing summer and winter bees, the unknown compound is by far the most noticeable. As confirmed in the PCA (Figure 2, Appendix A), the unknown compound, showing a singlet peak in the spectrum at 2.83 ppm is an important indicator of the distinction between the two groups. This metabolite has a particularly high concentration in the body of summer bees, while the presence is negligible in winter bees (Figure 3e, Table 1). A preliminary elucidation suggests that the signal belongs to *N*-methyl group of a non-proteinogenic amino acid and appears to be a powerful biomarker of seasonality or longevity. Structural elucidation is currently ongoing.

Concerning carbohydrates, winter bees had a significantly higher concentration of fructose, sucrose, and trehalose than in summer bees (Figure 3a, Table 1). Carbohydrates are a critical source of energy for bees, and are the exclusive fuel used for flight [39,40]. They can be converted into glycogen and stored in the fat body [41]. Higher carbohydrate intake was translated into higher glycogen levels and better diapause survival in reared bumble bees [42]. Trehalose is the primary blood sugar in most insects and is involved in multiple physiological roles [43]. One of which is a role as a stress protector. Studies of several insects have shown that stress conditions, such as drought, cold stress, and diapause, lead to trehalose accumulation in the body [44,45,46]. We observed increased levels of trehalose in the winter bees’ bodies (Figure 3a), this could be attributed to osmoregulation or enhancing cold stress tolerance in preparation for winter. Supporting this idea, an overwintering study of the mountain pine beetle (*Dendroctonus ponderosae*) using ^1^H NMR showed that the level of trehalose begins to increase from September until December as temperatures declined [47], a similar pattern has been shown in European spruce bark beetles (*Ips typographus*) [44], as well as in honey bees [48]. 

The standard deviation of sucrose, glucose, and fructose was higher than the mean value (Table 1). This is possibly originating from the crop leak during the sample disection or individual differences in sucrose feeding before the bees were harvested for this experiement. 

We observed that most free amino acids except for alanine and proline were significantly lower in winter compared to summer bees (Figure 3b). Although winter bees are characterized by generally higher protein content, vitellogenin concentration, and more antibacterial peptides than summer bees [16,21,22,33], their protein turnover is lower [49]. In contrast, the June samples were nurse bees, producing royal jelly proteins during the active brood rearing period [50,51], requiring a high protein synthesis and demand for free amino acids. 

Proline is the predominant amino acid in the hemolymph of honey bees [52,53] and is known as a cryoprotectant of insects, such as trehalose mentioned above [54,55]. In freeze tolerance studies in fruit flies (*Drosophila melanogaster*), proline and trehalose were the most abundant metabolites that accumulated during the cold assimilation [55,56,57]. We speculate that increased levels of trehalose and proline in the winter bees act as cryoprotectants that prevent membrane structure damages occuring during the cold winter, maintain water levels, or serve as an energy pool.

Environmental stressors such as pathogen exposure, starvation, and temperature fluctuation affect changes in insect metabolites. Winter bees emerge in autumn when brood rearing slows in the hive due to a drop in available pollen resources [15,58]. Such environmental changes could affect the metabolites in honey bees; this may be a part of their longevity mechanism. Low protein intake is known to be linked with lifespan extension [59]. A study found that honey bees kept on a diet low in essential amino acids showed longer lifespans through the Sir2 expression than those consuming a high essential amino acid diet [60]. Consistent with this result, Gomez-Moracho et al. [61] reported that insects fed a low-protein, high-carbohydrate diet showed the highest parasite prevalence but had the highest longevity. The decrease in amino acids in winter bees can be a contributing factor to the longevity mechanism in the long-living winter bees. Thus, future research focused on the metabolites correlation with the pathways that are known to regulate lifespan would expand the knowledge of honey bees’ longevity.

Vitellogenin is one of the most robust biochemical indicators in honey bees regarding their aging and longevity [18,20,62]. It is a phospholipoglycoprotein, a major precursor of egg-yolk protein synthesized by the fat body in female insects [20]. Vitellogenin contains phospholipids, mainly phosphatidylcholine and lysophosphatidylcholine [63,64]. In this study, we found significant changes in detected levels of choline and *O*-Phosphocholine in winter bees compared with summer bees (Figure 3c). Considering these two metabolites are the intermediates of phosphatidylcholine synthesis [65], we hypothesize that the changes in the choline-containing compounds from summer bees to winter bees could be related with the phosphatidylcholine synthesis as a vitellogenin component. 

The levels of adenosine monophosphate (AMP) and oxidized nicotinamide adenine dinucleotide (NAD^+^) have significantly increased in winter bees compared to summer bees (Figure 3d). Changes in AMP and NAD^+^ concentrations indicate cellular energy status and are meaningful for the study of longevity due to their correlation with the activation of AMP-activated protein kinase (AMPK) and sirtuins [66,67]. Sirtuins are NAD^+^-dependent protein deacylases. Increased NAD^+^ levels can enhance activated sirtuins that improve metabolic function and longevity [67]. AMP is one of the AMPK activators, an important energy sensor that is activated by increasing cellular AMP coupled with falling ATP [66]. Overexpression of AMPK has been shown to extend lifespan in *Caenorhabditis elegans* [68] and *Drosophila melanogaster* [69]. Previous honey bee studies have shown a lower concentration of NAD^+^ and decreased activity of AMPK and sirtuin in the abdomen of older bees compared to that of younger bees [70,71]. Therefore, we hypothesize that the significant increase in AMP and NAD^+^ in winter bees is one of the signals of switching to the long-living generation. Thus, further research is needed on this subject to more accruately link AMP and NAD^+^ with the regulation of longevity pathways.

For successful overwintering practices, we consider it is important to examine the metabolic changes from short-living summer bees to long-living winter bees. The molecular mechanisms and the detailed metabolic pathways regarding the changes in metabolites was not discerned during this experiment and require further research. The highlight of this study is that summer and winter bees have distinctive metabolic characteristics, and some metabolites could be useful biomarkers to distinguish these two groups.

## 5. Conclusions

This study observed and analyzed the metabolic differences between summer and winter bees of the same age. We identified and quantified a total of 36 metabolites, 28 metabolites which showed significant differences between the two groups.

Our results have the potential to provide a new analytical approach to confirm the presence of a long-living population in honey bee colonies. Furthermore, knowing robust biomarkers between summer and winter bees will guide future research on honey bee longevity and overwintering, and help to develop a practical overwintering strategy for the apiculture industry.

## Figures and Tables

**Figure 1 insects-13-00193-f001:**
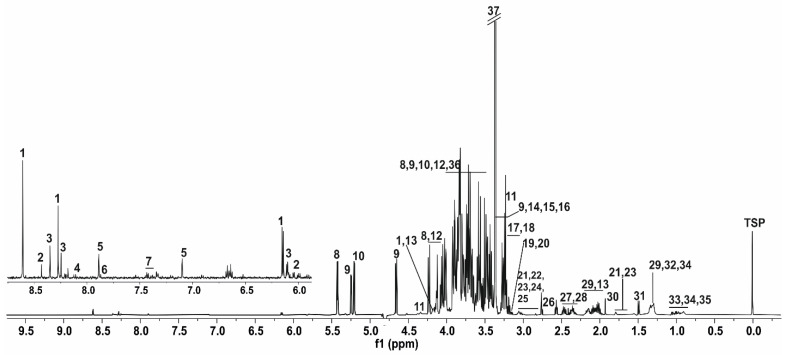
Typical proton nuclear magnetic resonance (^1^H NMR) spectrum of honey bee worker body (gut and venom sac excluded). The numbers correspond to these quantified compounds: 1. Adenosine monophosphate (AMP), 2. Oxidized nicotinamide adenine dinucleotide (NAD^+^), 3. Inosine, 4. Adenine, 5. Histidine, 6. Uridine, 7. Phenylalanine, 8. Sucrose, 9. Glucose, 10. Trehalose, 11. *sn*-glycero-3-phosphocholine, 12. Fructose, 13. Proline, 14. *O*-phosphocholine, 15. Taurine, 16. Trimethylamine *N*-oxide, 17. Choline, 18. β-alanine, 19. Dimethyl sulfone, 20. Malonate, 21. Putrescine, 22. Creatine, 23. Lysine, 24. Asparagine, 25. Unknown, 26. Sarcosine, 27. Glutamine, 28. Succinate, 29. Suberate, 30. Acetate, 31. Alanine, 32. Threonine, 33. Valine, 34. Isoleucine, 35. Leucine, 36. Glycine, and 37. Methanol.

**Figure 2 insects-13-00193-f002:**
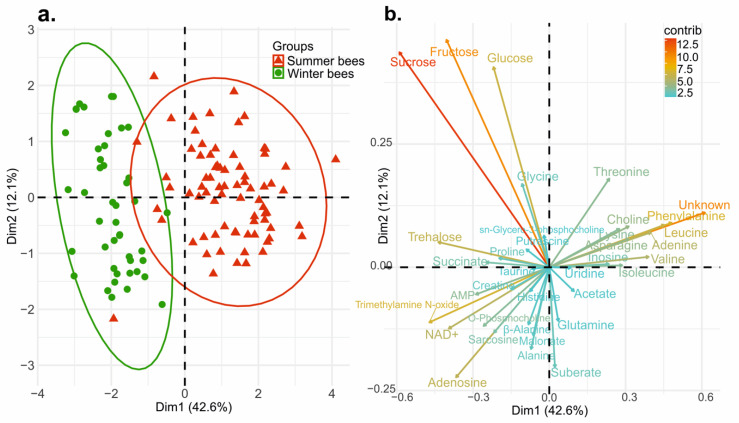
Principal component analysis of summer and winter bees (**a**) Score plot. Dim1(PC1) on the *x*-axis explains 42.6%, Dim2 (PC2) on the *y*-axis explains 12.1% of variability between the summer bees and winter bees; (**b**) Variable correlation plot. The length and colors of the arrows explain how each variable expresses the differences between the two compared groups (summer and winter). The longer the arrow, the stronger its influences in discrimination. The red and orange colors show a high contribution on each principal component, whereas the green and blue show a relatively low contribution.

**Figure 3 insects-13-00193-f003:**
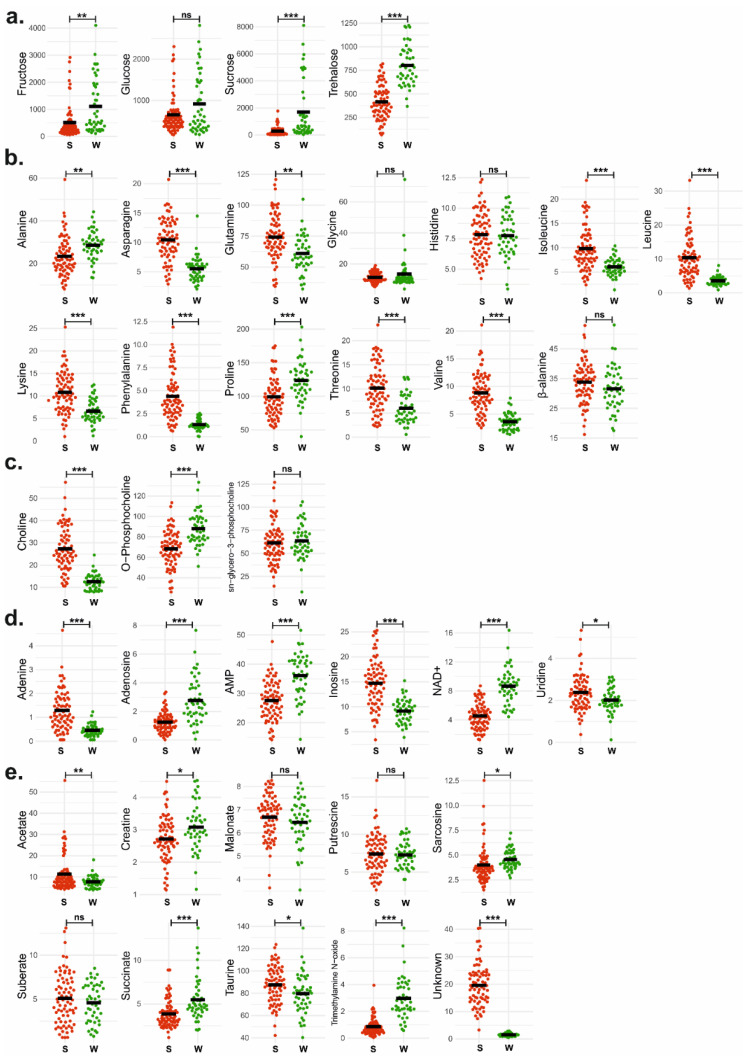
Comparison of 36 metabolites between summer bees and winter bees: (**a**) Carbohydrates; (**b**) Amino acids; (**c**) Choline-containing compounds; (**d**) Nucleoside, nucleotides, and nucleobases; (**e**) Others. *X*-axis: the groups summer (red) and winter (green), *y*-axis: metabolite concentration (μg/bee), black bar: mean. The P-values were obtained from the independent *t*-test. Asterisk indicates significant differences * *p* < 0.05, ** *p* < 0.005, *** *p* < 0.0001, ns = not significant. Benjamini–Hochberg’s procedure (false discovery rate, FDR 0.05) is applied after the independent *t*-test. Details of the independent *t*-test and Benjamini–Hochberg’s procedure informed in Table 1.

**Table 1 insects-13-00193-t001:** Mean (μg/bee), standard deviation (SD), independent *t*-test, and Benjamini–Hochberg’s procedure (FDR 0.05).

	Summber Bees (*n* = 75)		Winter Bees (*n* = 45)			Independent *t*-Test	Benjamini–Hochberg
Metabolite	Mean	SD	Mean	SD	DF ^1^	*p*-Value	*p*-Value
AMP	27.56	6.60	36.12	7.55	118	0.0019	<0.0001
Acetate	11.29	8.42	7.71	2.69	118	<0.0001	0.0027
Adenine	1.29	0.82	0.46	0.25	118	<0.0001	<0.0001
Adenosine	1.25	0.68	2.76	1.58	118	0.0001	<0.0001
Alanine	23.38	8.90	28.56	6.72	118 [115.6]	<0.0001	0.0001
Asparagine	10.45	3.55	5.52	2.09	118	<0.0001	<0.0001
Choline	27.35	9.66	12.57	3.50	118	<0.0001	<0.0001
Creatine	2.72	0.71	3.08	0.72	118	0.0120	0.0166
Fructose	509.4	611.6	1109.1	1008.3	118 [84.2]	0.0001	0.0001
Glucose	655.8	442.2	915.0	715.6	118 [68.8]	0.1664	0.1997
Glutamine	74.25	18.29	60.93	14.42	118	0.0001	0.0002
Glycine	11.37	3.12	13.52	11.00	118	0.3608	0.4059
Histidine	7.82	1.85	7.75	1.73	118	0.8449	0.8449
Inosine	14.73	4.76	9.17	2.36	118	<0.0001	<0.0001
Isoleucine	9.82	4.41	6.06	1.77	118 [112.8]	<0.0001	<0.0001
Leucine	10.35	6.02	3.54	1.31	118	<0.0001	<0.0001
Lysine	10.83	4.45	6.67	2.59	118	<0.0001	<0.0001
Malonate	6.68	0.93	6.46	0.97	118	0.2347	0.2726
NAD^+^	4.54	1.78	8.66	2.48	118 [116.8]	<0.0001	<0.0001
*O*-Phosphocholine	68.38	17.75	88.01	15.99	118 [117.8]	<0.0001	<0.0001
Phenylalanine	4.39	2.53	1.31	0.55	118	<0.0001	<0.0001
Proline	99.24	29.01	123.47	29.05	118	<0.0001	<0.0001
Putrescine	7.41	2.62	7.28	1.75	118 [116.6]	0.7555	0.7771
Sarcosine	4.00	1.80	4.55	0.97	118 [117.5]	0.0008	0.0012
Suberate	5.10	2.90	4.58	2.19	118	0.5916	0.7204
Succinate	3.86	1.53	5.49	2.47	118	0.6804	<0.0001
Sucrose	284.3	301.9	1690.3	2135.0	118	<0.0001	<0.0001
Taurine	87.57	16.47	79.62	18.12	118	<0.0001	0.0167
Threonine	10.19	4.73	6.00	2.97	118	0.0125	<0.0001
Trehalose	416.0	181.1	801.8	212.0	118 [115.7]	<0.0001	<0.0001
Trimethylamine *N*-oxide	0.86	0.63	2.97	1.54	118	<0.0001	<0.0001
Unknown	19.61	7.62	1.53	0.55	118	<0.0001	<0.0001
Uridine	2.38	0.85	2.01	0.59	118	<0.0001	0.0499
Valine	8.83	3.60	3.61	1.52	118	0.0388	<0.0001
*sn*-Glycero-3-phosphocholine	61.28	21.28	63.42	18.68	118	<0.0001	0.6454
*β*-Alanine	33.81	6.80	31.56	7.31	118	0.0761	0.0945

^1^ Correction to the degree of freedom (DF) has applied when the equal variance is not assumed and the corrected value is shown in brackets.

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
