# Peer review of "^1^H NMR Profiling of Honey Bee Bodies Revealed Metabolic Differences between Summer and Winter Bees"

_insects, 2022, doi:10.3390/insects13020193_

Round 1

Reviewer 1 Report

We know little about what defines a winter bee relative to a summer bee. While these differences are inherently interesting, we can apply this topic to reduce overwinter colony loss, which is a big determinant of annual hive losses.

In this paper by Lee et al., the authors looked at the differences between summer and winter bees. Summer bees were collected in June and winter bees were collected in August. Bees were samples from three hives and were all 11d of age. The authors used NMR to look at the metabolites present in the whole body extracts of these summer and winter bees. Principal components analysis showed differences in the summer and winter bees and further analyses determined what metabolites were responsible for the differences.

I liked this paper. It is well written and will be a nice addition to the literature, which is fairly scant at this point. I have few negative comments. My only major concern is in how the authors are defining a winter bee. This could use more explanation in the methods and perhaps even the discussion. They collected bees from the hives in August and make the assumption that these are, in fact, winter bees based on what they say is reduced brood area in the hives. Can the authors back up their conjecture that these are, in fact, winter bees? In other words, if they were not destructively sampled for NMR, would you expect that each of these bees would have lived for 140+ days? Obviously, it’s impossible to estimate the lifespan of a bee that you crushed up for NMR, but what about the others in their cohort? Or what is the probability that you would have found a long-lived bee from this August cohort? Could you have used part of the bee for gene expression?

I recommend that the paper goes back to the authors requesting revisions. I attached the annotated .pdf that describes what needs to be addressed.

Specific comments:

Line 21 – to the different lifespans

Line 40-41 – clarify which season you are referring to

Line 61 – “the one of the most well-known” – take out first “the”

Line 72 – opposite pattern

Line 77 – “shown a greater dry weight” – take out “shown”

Line 78 – “Also, they have shown” – take out “shown”

Line 115 - It is important to clarify somewhere that you did not explicitly determine that these were long-lived winter bees. This would require one to determine the lifespan of other bees in the marked cohort or to validate this with other markers like vg titer. Please address this here or in the discussion.

Line 118 - For your stats below (line 159), can you clarify whether you used hive as the replicate? Or did you use the individual as the replicate? If the latter, would hive be a random effect?

Line 126 – specify that you removed these tissues.

Line 127 - How many bees per hive did you analyze? Also, did you include the entire bee (head, thorax, abdomen) in your sample for NMR? If so, it's interesting to think about the differences coming not only from the fat body but also the head (if these bees were nursing).

Line 151 – for the compounds below the limit of detection – is it standard practice to replace these values with those that are 1/3 the lowest value in the data set? This could mess up your stats and make the data skewed. And also you are inputting what is basically a made-up value, which doesn’t seem right.

Line 153 – Did you weigh each bee? Dry weight or wet weight? If not, what is the point of using one number, other than to force the data into a mass per unit of bee measurement.

Line 199 – the OPLS-DA - is this described in your methods? or is a part of the analysis that you already mentioned?

Line 229 – Do you mean carboxylic acids and amines?

Line 267 – move the comma after compound to after ppm

Line 273 - Have you seen the paper by Woodard et al. 2019 (https://doi.org/10.1093/conphys/coz048) where they find that carbohydrates are important for bumblebee diapause? This could apply to your experiment, if you believe that bumblebee could be similar to honey bee wintering.

Line 287 – This paragraph would be better in the results.

Line 292 – Combine the first and second sentences in the paragraph

Line 300 - Were the bees that you collected at 11d of age were foraging or were they nursing brood? If they were in the hive, I suspect that the demands were mostly from brood rearing, since a lot of protein is produced to make jelly.

Line 301 - It's interesting that proline was higher in the winter bees because fall pollens are lower in proline than spring pollens, at least in Arizona, USA. (see https://doi.org/10.3390/insects12030235)

Line 309 - This paragraph should be revised. Are you talking about stress or how a drop in pollen resources could trigger winter bee production and, further, the change in metabolites in winter bees could influence their longevity. I would go with the latter and take out the sentences in lines 309-312. Although it is tempting to talk about the drop in pollen as a nutritional stress, I would simply say that it's a natural drop in pollen resources and that is correlated with winter bee production and reduced brood rearing. It's only a correlate at this point and we don't know whether the drop in pollen causes winter bee production. (Although it likely causes the differences in brood rearing.)

Throughout, but especially lines 310 and 311 – Take out language like “were reported to have” and “has been shown to increase” and simply say have or increase.

Line 313 – nutritional is misspelled

Line 318 – Consistent

Line 332 - Does it make sense that choline is higher in summer bees but O-phosphocholine is higher in winter bees? Choline is a micronutrient, not a lipid. So does choline fall within your vg hypothesis. I believe that O-phosphocholine does, but not choline.

Line 346 – do not capitalize sirtuin?

Line 354 – require further research because it’s referring to two things that require further research

Line 359 – This paragraph is repetitive and should be deleted or incorporated into the first paragraph of the discussion.

Line 373 - Soften this language because you did not explicitly show that these are winter bees. All you know is that this is probably when the hive produces winter bees (see above comment).

Author Response

Dear Reviewer,

Thank you for a detailed review. We have edited the manuscript following your comments and detailed replies are hereby attached.

Reviewer 2 Report

See attached file

Author Response

Dear Reviewer,

Thank you for spending your time to read this manuscript carefully. We deeply appreciate your further suggestions and specific comments on the manuscript.

We have edited manuscript following your comments and detailed replies are attached.

Round 2

Reviewer 2 Report

The authors fixed some of the minor problems I had with this manuscript. I still think that “Is there any metabolic differences between the summer and winter bees?” Is a kind of oversimplified research question and the authors not really give a good answer for what is the difference in metabolism between the two bee morphs.

I do not have any additional specific comments.